# FAIRDROPOUT: USING EXAMPLE-TIED DROPOUT TO ENHANCE GENERALIZATION FOR MINORITY GROUPS

## ABSTRACT

Deep learning models frequently exploit spurious features in training data to achieve low training error, often resulting in poor generalization when faced with shifted testing distributions. To address this issue, various methods from imbalanced learning, representation learning, and classifier recalibration have been proposed to enhance the robustness of deep neural networks against spurious correlations. In this paper, we observe that models trained with empirical risk minimization tend to generalize well for examples from the majority groups while memorizing instances from minority groups. Building on recent findings that show memorization can be localized to a limited number of neurons, we apply example-tied dropout as a method we term *FairDropout*, aimed at redirecting this memorization to specific neurons that we subsequently drop out during inference. We empirically evaluate FairDropout using the subpopulation benchmark suite encompassing vision, language, and healthcare tasks, demonstrating that it significantly reduces reliance on spurious correlations.

## 1 INTRODUCTION

Deep neural networks trained with empirical risk minimization (ERM) continue to achieve remarkable performance on a wide range of tasks. However, ERM-trained models can experience a drop in predictive performance when facing a variety of subpopulation shifts (Yang et al., 2023; Quionero-Candela et al., 2009). In particular, if datasets contain spurious features, i.e., patterns that are highly predictive of the training labels but not causally related to the target, ERM may fail to learn robust features that generalize across subpopulation shifts (Geirhos et al., 2020). For example, image classifiers can make use of non-robust features such as image backgrounds or hair colors, which may be not relevant to the task. The usage of such spurious features (e.g., hair colors) for some domains can hurt the *fairness* of classifiers, thus raising potential safety concerns in deployment (Amodei et al., 2016).

To address the problem of learning more robust features in the presence of spurious features, several works have been proposed. The widely practical setup is to assume the presence of a group partition in datasets (Liu et al., 2021; Sagawa et al., 2019). In such a setting, training labels and spurious features can be highly correlated in a particular group of the training distribution, but not in testing distributions. Thus, naive training algorithms can easily maximize training performance by relying on spurious features, but observe a significant drop in worst-group performance on testing when this correlation does not hold. Most of the existing work in this area assumes the presence of group annotations in the training set to learn more robust-to-spurious-correlation features. For example, GroupDRO (Sagawa et al., 2019) directly minimizes the worst group error. However, this type of work has a hard requirement to know prior training group labels, which is impractical in large-scale datasets. There exist other works that do not assume this availability of group annotations on the training set. An example is DFR Kirichenko et al. (2023), which observes that ERM learns *core* (or robust) and spurious features, and then proposes a two-stage approach, where the first stage is ERM and the second stage is classifier retraining with a group-balanced validation dataset. While DFR has been successful in improving worst group performance, it still needs group annotations to form a group-balanced set to down-weigh spurious features.

In this work, we hypothesize that reducing example-level memorization can address spurious correlations without the need for group labels. Building on recent advances that explain the interconnec-

tion between memorization and generalization (Baldock et al., 2021; Maini et al., 2023), we study this interconnection for the first time in the context of spurious correlation. We further apply a recently proposed technique to localize memorization, originally designed in label noise settings with small networks, scaling it to larger networks, and demonstrating its application to the spurious correlation setting for the first time. We name our method *FairDropout*. FairDropout fairly distributes *memorizing neurons* during training, and during dropout, we drop out these neurons. Our contributions can be summarized as follows: (i) We show a discrepancy in behaviors between majority group and minority group generalization and link this phenomenon to memorization. (ii) We show for the first time that one can scale and apply the example-tied dropout –previously used in *label noise* settings with smaller networks (Maini et al., 2023)– to larger architectures such as ResNet-50 (He et al., 2016) and BERT (Sung et al., 2019), and term it as *FairDropout*. (iii) We evaluate FairDropout on the subpopulation benchmark suite and show improvements over worst-group accuracy on image, medical (X-Ray), and language tasks.

## 2 RELATED WORK

Methods have been proposed to fight against spurious correlation.

### 2.1 USING TRAINING GROUP INFORMATION

Most methods that fight against spurious correlation assume to have training group annotations. Some methods directly adapt ERM. For instance, groupDRO (Sagawa et al., 2019) and its variant CVaRDRO (Duchi & Namkoong, 2021) aim to minimize the worst group error rather than the average error used by ERM. Similarly, when group information is known, methods from out-of-distribution generalization (Arjovsky et al., 2019; Krueger et al., 2021; Wald et al., 2021; Krueger et al., 2021) can be framed to learn more robust-to-spurious-correlation features. Other approaches use training group information to synthetically augment minority group samples via generative modeling (Goel et al., 2020). Reweighting and subsampling techniques can also be employed to balance majority and minority groups (Sagawa et al., 2020; Byrd & Lipton, 2019). However, all these works share a major limitation: they rely on the knowledge of group information, which is not easily scalable to large datasets. Manually annotating group labels requires task-specific expertise, making it prohibitively expensive.

### 2.2 WITHOUT USING TRAINING GROUP INFORMATION

Given the expensive cost of manual group annotation, there has been a growing interest in combating spurious correlation without group annotations in the training set. Some methods, after observing the training dynamics of SGD, propose regularization terms based on margins to learn more robust features (Pezeshki et al., 2021; Puli et al., 2023). Two-stage algorithms, among the most popular methods that do not assume the knowledge of training group information, typically start with ERM. In this first stage, the minority group is inferred, and in the second stage, robustness to spurious correlations is introduced, for example through contrastive learning (Zhang et al., 2022) or by upweighting the loss of inferred minority group samples (Qiu et al., 2023; Liu et al., 2021). A recent study (Kirichenko et al., 2023) explains the importance of the first stage ERM, by showing that ERM learns both spurious and *core* (or robust to spurious correlation) features. It then proposes retraining only the classifier head in the second stage using a group-balanced validation set. This approach has been extended to HTT-DFR (Hameed et al., 2024), where the second phase involves retraining a sparse network. Our work is inspired by this observation of the ability to learn core features from ERM, but instead of using a more computationally demanding two-phase algorithm, our work on reducing spurious features by its link to the memorizing neurons.

### 2.3 MEMORIZATION AND GENERALIZATION LINKS

There have been recent advances in exploring and explaining the links between generalization and memorization. Memorization is seen here as the ability to correctly predict *atypical* examples with potentially wrong patterns (Maini et al., 2023). In particular, Jiang et al. (2021); Carlini et al. (2019) have developed metrics to quantify to which extent an example is regular or atypical. Some works firstly have established that memorization happens in later layers Baldock et al. (2021); Stephenson

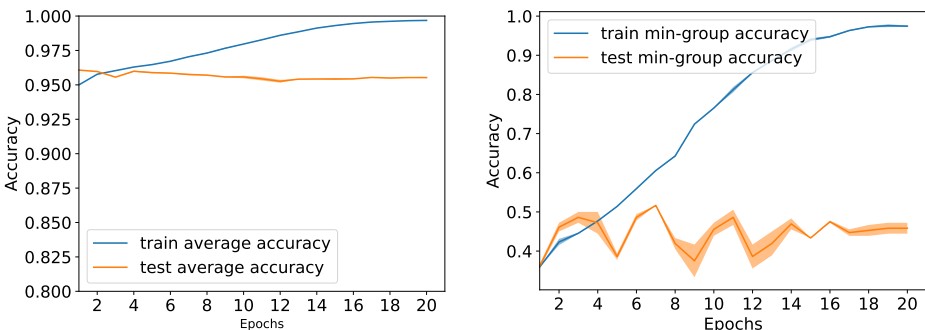

Figure 1: Discrepancy in generalization behaviors between majority groups and minority groups on CelebA. We observe that models trained exhibit a large generalization gap on minority groups, a synonym of minority group overfitting.

et al. (2021) while a Maini et al. (2023) recently show that it can appear at every network depth. Furthermore, Maini et al. (2023) conceptualizes the idea of localizing memorization by computing the minimum number of neurons required to flip predictions. This paper uses the Maini et al. (2023)'s method to flag memorization not in the context of label noise as them but in the context of spurious correlation. Indeed, there have been various works that show that when mechanistically interpreting deep neural networks, there are neurons that specialize for certain tasks Zenke et al. (2017); Cheung et al. (2019); Hendel et al. (2023).

## 3 METHODS

This section begins by outlining the problem, followed by an analysis of minority group example memorization, and then introduces FairDropout.

### 3.1 PROBLEM DESCRIPTION

We consider a classification problem with a training sample $\mathcal{D}_{\text{tr}} = \{(\boldsymbol{x}, y)\}_{i=1}^{N}$ drawn from a training distribution $p_{\text{tr}}$, where $\boldsymbol{x}_i \in \mathcal{X}$ is the input and $y_i \in \mathcal{Y}$ is its class label. We further assume that the existence of a spurious attribute $a \in \mathcal{A}$, which is non-predictive of $y$ Ye et al. (2024). We denote by groups, the pairs $g := (y, a) \in \mathcal{Y} \times \mathcal{A} := \mathcal{G}$. Since $a$ is not predictive in $y$, if there is a correlation between $y$ and $a$ in the training distribution $p_t r$, and not in the test distribution $p_{te}$, therefore trained models may performance drop in groups where this correlation does not hold.

For example, CelebA (Liu et al., 2015) is one of the most popular datasets in the spurious correlation literature. The common task is to predict the hair color in celebrity faces ($\mathcal{Y} = \{\text{blond hair}, \text{non-blond hair}\}$), and the spurious attribute is the gender ($a \in \{\text{woman}, \text{man}\}$). In the CelebA training set, only $1\%$ of faces are the group from blond men. Therefore, trained models may rely on the spurious gender feature to determine hair color. Therefore, when evaluating the predictive performance, one might not simply assess the average testing performance; the worst-group accuracy, such as accuracy on blond men, may become crucial.

Formally, considering a parameterized model $f_{\boldsymbol{\theta}} : \mathcal{X} \longrightarrow \mathcal{Y}$, the goal of learning in the presence of spurious correlation is to find the model that will minimize the worst-group expected error

$$\max_{g \in \mathcal{G}} \quad \mathbb{E}\left[\ell_{0-1}\left(f_{\boldsymbol{\theta}}\left(\boldsymbol{x}\right), y\right) | g\right], \tag{1}$$

where $\ell_{0-1}\left(f_{\boldsymbol{\theta}}(\boldsymbol{x}), y\right) = \mathbf{1}\left[f_{\boldsymbol{\theta}}(\boldsymbol{x}) \neq y\right]$ is the 0-1 loss (Liu et al., 2021). We are interested in the case where there no available group information for the training set, but it is only accessible during testing for evaluation.

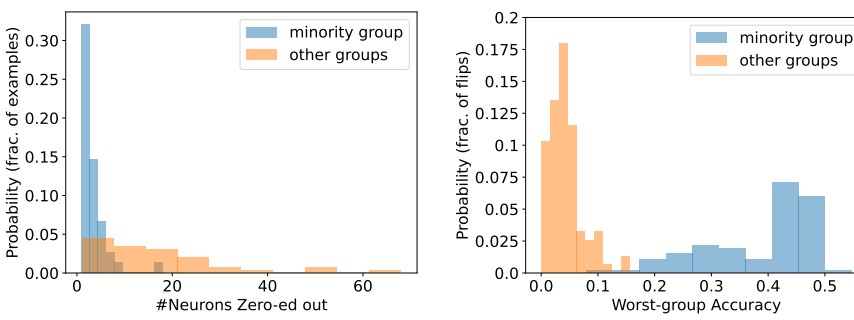

(a) Number of Neurons to Flip Predictions.    (b) Impact on Training Worst-group Accuracy.

Figure 2: For each example in a subset of 100 from the minority group and 100 from other groups, we iteratively remove the most important neurons from a ResNet-50 model trained on the CelebA dataset, until the example's prediction flips. (a) Minority-group examples need fewer neurons to flip their prediction. (b) After dropping these neurons to flip the prediction for each example, the drop in worst-group accuracy is greater from the flip related to majority groups than for the minority group, indicating that minority-group examples are being memorized.

### 3.2 MEMORIZATION OF MINORITY GROUP EXAMPLES

We start our analyses with ERM on CelebA, which is the most popular, real-world, and large dataset for studying spurious correlation, making it generalizable to real-world settings. As explained earlier, it has 4 groups, namely from {blond hair, non-blond hair} × {woman, man}, with the minority group being (blond, man). We train ResNet-50 (He et al., 2016) with ERM and track train/test performance on the different groups.

Fig. 2 shows the average and worst group performance. It can be observed in Fig. 2, a discrepancy in generalization behaviors between the majority groups (represented by the average performance, but the same behavior applies) and the minority group. Specifically, we observe a *large generalization gap for the minority group*, a synonym of overfitting–a point that has not been sufficiently emphasized in prior research.

We now turn our attention to analyzing the underlying causes of this failure in minority group generalization through the lens of memorization. Indeed, deep neural networks are high-capacity models capable of fitting complex and atypical examples, making it reasonable to associate this generalization failure with memorization. Leveraging recent advances in understanding memorization, we employ the method recently proposed by Maini et al. (2023) to detect memorization. This technique consists in finding the minimum number of neurons (channels for the case of convolutional layers) materialized by $z^{(l,j)}$ ($l$ being layer indexes, and $j$ being neuron indexes) that preserve the training sample's prediction while maximizing the loss on the input to whose prediction should be flipped. For an input $\boldsymbol{x}_i$, this is technically done by sequentially computing for each iteration

$$z_*^{(l,j)} = \arg\max_l \left[ \nabla_{\boldsymbol{\theta}_l} \left( \mathcal{L}\left( f_{\hat{\boldsymbol{\theta}}}(\boldsymbol{x}_i), y_i \right) - \frac{1}{|\mathcal{B}|} \sum_{(\boldsymbol{x},y) \in \mathcal{B}} \mathcal{L}\left( f_{\hat{\boldsymbol{\theta}}}(\boldsymbol{x}), y \right) \right) \right]_j, \qquad (2)$$

where $\mathcal{B}$ is the random batch on which the predictions have to be conserved, $f_{\hat{\boldsymbol{\theta}}}$ is the current iteration of the modified model (model on which a neuron was dropped in the previous iteration). The sequential procedure continues until the prediction of $\boldsymbol{x}_i$ is flipped. The final neuron indexes $j$ are seen as the most critical neurons that are only related to the considered example, and Maini et al. (2023) show that the proportion of these neurons can be used to detect memorized examples.

We conduct this experiment to analyze the memorization behaviors in the context of spurious correlation on the CelebA dataset. Fig. 2 shows the number of neurons required to flip prediction each prediction on a subsample of majority and minority groups. We observe that in general, (i) the number of neurons required to flip each prediction from the minority group is considerably lower than the corresponding number for majority groups (see the left-most of the plot of Fig 2a). Furthermore, as shown in Fig. 2b, (ii) these neurons have even less effect on training worst-group accuracy than

the corresponding ones from the majority group. Referring to a similar analysis from Maini et al. (2023), (i) and (ii) indicate that minority group examples are more prone to memorization issues.

While we have identified generalization problems and memorization issues within the minority group, there is no direct evidence suggesting a causal link between the two phenomena in this context of spurious correlation. To investigate this potential link, we conducted a new experiment to measure changes in test worst-group accuracy after dropping out neurons from the minority groups.

Fig. 3 shows the test worst-group accuracy after individually (per example) dropping neurons. We observe that for approximately 75% of the drops, the worst-group accuracy significantly improves. This means that most large proportion of these memorizing neurons are detrimental to minority group generalization.

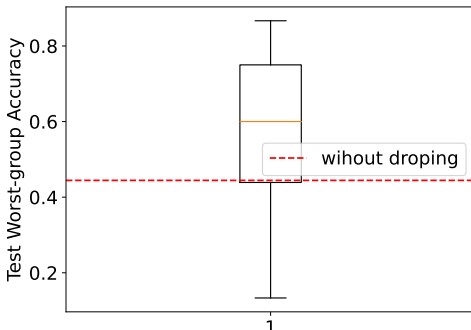

Figure 3: Effect on the worst-group accuracy when dropping memorizing neurons as shown in 2. For each example in the minority-group sample, we drop the minimum number of neurons to flip its prediction. From the quartiles on this figure, we observe that in ≈ 75% of cases dropping out *memorizing* neurons significantly improves test worst-group accuracy.

### 3.3 THE EXAMPLETIEDDROPOUT AS A FAIRDROPOUT

After observing that certain neurons are closely tied to specific examples (particularly the memorizing ones in the minority group) and that dropping them positively impacts minority group generalization, it becomes important to crucial to leverage this insight by directing this memorization to fixed neurons. Drawing inspiration from the example-tied dropout introduced in the context of label noise by Maini et al. (2023) for small networks such as ResNet-9, and smaller datasets such as MNIST (Deng, 2012) or CIFAR-10 (Krizhevsky, 2009), we introduce the FairDropout, which is an example-tied dropout in the context of spurious correlation. Unlike the original example-tied dropout, FairDropout can be applied not only after any intermediate layers but also after any newly added projection layer before the linear head.

As an example-tied dropout, the FairDropout is a layer without learnable parameters, that divides neurons into two types, governed by two hyper-parameters: $p_{\text{gen}}$ and $p_{\text{mem}}$. The first set of neurons are the generalizing neurons, which are seen by every example in the dataset. If the preceding layer of the FairDropout has the size $H$, then there are $p_{\text{gen}}H$ neurons are designated as generalizing. The remaining of $(1 - p_{\text{gen}})H$ neurons are memorizing neurons and each sample is allocated a memorizing neuron uniformly with probability $p_{\text{mem}}$. Furthermore, the *fair* prefix comes from the fact that every example allocates the same fixed number of memorizing neurons. As depicted in Fig. 5, during training, given an example, the FairDropout propagates its generalizing features and its example-wise memorizing ones. In this case, each image allocates only one memorizing neuron. During testing, the memorizing ones are dropped. Finally, we observe that when $p_{\text{gen}} = 1$ the FairDropout is just an identity function and trained models correspond to ERM-trained models.

## 4 EXPERIMENTAL RESULTS

We conduct experiments to evaluate the FairDropout on CelebA as sanity check and on a benchmark suite.

### 4.1 WARM-UP ON CELEBA: FAIRDROPOUT BALANCES GROUP ACCURACY

We incorporate the FairDropout after the third residual block on ResNet-50, with the hyperparameters $p_{\text{mem}} = p_{\text{gen}} = 0.2$, and track the train/test average/worst-group accuracy.

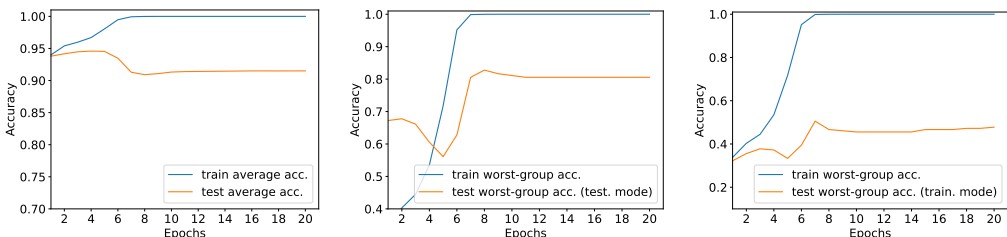

Figure 4: Training with FairDropout on CelebA. Train/test average and worst-group accuracy with FairDropout are plotted. Training and testing mode respectively refer to the evaluation without dropping memorizing neurons, and after dropping them. We observe that dropping out these memorizing neurons has the benefit of improving worst-group accuracy.

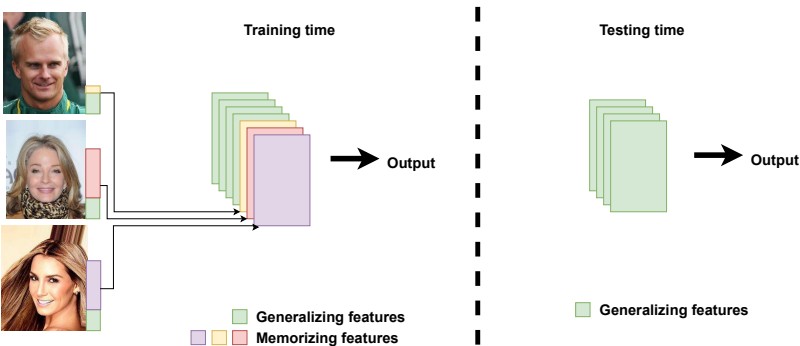

Figure 5: Example-Tied Dropout as a FairDropout. The FairDropout redirects the example memorization on specific neurons. Memorizing neurons are uniformly allocated to training examples during training. During testing, these features are dropped.

Fig. 4 shows the evolution of the train/test average and worst-group accuracy throughout epochs. It can be observed on the right-most plot that if using the training mode (in this mode, memorizing neurons are kept) of the FairDropout to evaluate accuracy, we obtain almost the same behavior as if there was no FairDropout, i.e., ERM. Indeed, the worst-group accuracy saturates around $45\%$ as in Fig. 1.

In contrast, in the testing mode (in this mode, memorizing neurons are dropped) of the FairDropout, the worst-group accuracy does not saturate around $45\%$, but around $80\%$. Thus dropping out memorizing neurons after training with the FairDropout has a clear effect on boosting the worst-group accuracy. In the following section, we compare the FairDropout against state-of-the-art methods in spurious correlation.

## 4.2 BENCHMARKING FAIRDROPOUT WITH BASELINES

Before presenting the results of the comparison between baselines, we present the experimental setup used largely inspired from Yang et al. (2023).

### 4.2.1 EXPERIMENTAL SETUP

We use the recently proposed subpopulation shift library and benchmark suite (Yang et al., 2023) that implements the state-of-the-art methods in spurious correlation [1].

We use 5 diverse datasets that are very used in spurious correlation litterature (Yang et al., 2023).
**Waterbirds.** Waterbirds Wah et al. (2011). Waterbirds is a well-known *synthetic* image dataset for binary classification. The task is to classify whether a bird is a landbird or a waterbird. The

---

[1]Code is available at this ANONYMOUS LINK.

spurious attribute is the background (water or land). There are therefore 4 groups that are from {landbird, waterbird} × {water background, land background}.

**CelebA.** As introduced in Sec. 3.1, CelebA (Liu et al., 2015) is one of the largest, real-world image datasets used in the context of spurious correlation. It has around 200,000 celebrity face images. The task, in the spurious correlations literature, is to predict the hair color of persons (blond vs. non-blond) and the spurious correlation is the gender.

**MetaShift.** The dataset Metashift (Liang & Zou, 2022) that we use here is an image dataset that was built by Yang et al. (2023). The goal is to distinguish between the two animals (cats vs dogs). The spurious attribute is the image background. Cats are more likely to be indoors, while dogs are more likely to be outdoors.

**MultiNLI.** The MultiNLI dataset (Liang & Zou, 2022) is a text dataset very used in spurious correlation literature. The target is the natural language relationship between the premise and the hypothesis. It has three classes (neutral, contradiction, or entailment). The spurious attribute is a variable that tells whether negation appears in the text or not. Indeed, negation is highly correlated with the contradiction label. **MIMIC-CXR.** MIMIC-CXR (Johnson et al., 2019) is a chest X-ray dataset, where its approximately 300,000 images come from the Beth Israel Deaconess Medical Center from Boston, Massachusetts. We use the setting of Yang et al. (2023), where the label is "No Finding" as the label. The positive class means that the patient is not ill. the spurious attribute domain is the cross-product of race (White, Black, Other) and gender.

All the data preprocessing and train/val/test splits are directly adopted from Yang et al. (2023) as we implement our method in their library.

**Models.** As in the benchmark (Yang et al., 2023), we use the Pytorch pretrained ResNet-50 models for image datasets and BERT Sung et al. (2019) for the MultiNLI text datasets.

**Metrics.** According to most previous works, we evaluate the reliance on spurious correlation through worst-group accuracy.

**Baseline methods.** We compare the FairDropout with state-of-the-art algorithms implemented in the subpopulation shift benchmark. Our work does not need the knowledge of group information. We thus evaluate our method in the setting where we do not have group information. However, methods that need group information have been converted by Yang et al. (2023) to an equivalent method by considering class information instead of group. For example, GroupDRO can be converted by an equivalent goal of minimizing worst-class error. Benchmarked methods from the spurious correlation literature include GroupDRO (Sagawa et al., 2019), CVaRDRO (Duchi & Namkoong, 2021), JTT (Liu et al., 2021), LfF (Nam et al., 2020), LISA . There are also two-phase methods that retrain the classifier, which are DFR (Yao et al., 2022) (retraining is done on the *validation* set), CRT and its variant ReWeightCRT (Kang et al., 2020). Finally, we also include methods that are mostly designed for the imbalanced learning problem, which are ReSample (Japkowicz, 2000), ReWeight (Japkowicz, 2000), SqrtReWeight (Japkowicz, 2000), CBLoss (Cui et al., 2019), LDAM Cao et al. (2019) and BSoftmax (Ren et al., 2020). Note that the FairDropout technique can be combined with any of these baseline methods to boost its performance.

**Hyperparameter tuning.** As we consider the most difficult case we do not have group information for the training and validation sets, similarly with the benchmark Yang et al. (2023), we tune the $p_{\text{mem}}, p_{\text{gen}}$, learning rate, and weight decay with the worst-class accuracy. We use the SGD optimizer with weight decay.

**Positions of the FairDropout Layers.** In principle, the FairDropout layer can be placed after any intermediate layer in the network. However, in large-scale, potentially pre-trained models, the placement of FairDropout may require careful consideration. In models with skipped connections as in ResNet-50, in our settings, we consider possible positions after residual blocks. In BERT-like models, we propose adding a new linear layer before the classifier head and positioning the FairDropout layer there. This ensures that the pertaining features are preserved while controlling memorization during fine-tuning. The optimal placement, however, depends on the dataset, as spurious correlations exhibit task-specific levels of abstraction. Therefore, we tune the position of FairDropout along with other optimization hyperparameters using worst-class accuracy as a guiding metric.

Table 1: Comparison of the FairDropout against state-of-the-art methods when spurious attribute annotations or group annotations are unknown in both train and validation. Test worst-group accuracy is reported and is obtained from the subpopulation shift benchmark Yang et al. (2023). The symbol ○ indicates that the original method requires group information for the training whereas ● means that it requires group information for the validation.

| Method Types | Algorithm | Waterbirds | CelebA | MetaShift | MultiNLI | MIMIC-CXR |
|---|---|---|---|---|---|---|
| standard | ERM | 69.1 ±4.7 | 57.6 ±0.8 | 82.1 ±0.8 | 66.4 ±2.3 | 68.6 ±0.2 |
| Data augmentation | Mixup | 77.5 ±0.7 | 57.8 ±0.8 | 79.0 ±0.8 | 66.8 ±0.3 | 66.8 ±0.6 |
| Spurious correlation | **FairDropout (ours)** | 70.6 ± 0.2 | **75.6 ± 2.1** | **85.9 ± 1.1** | **70.3 ± 2.4** | **70.6 ±0.6** |
| | ○ GroupDRO | 73.1 ±0.4 | 68.3 ±0.9 | 83.1 ±0.7 | 64.1 ±0.8 | 67.4 ±0.5 |
| | ○ CVaRDRO | 75.5 ±2.2 | 60.2 ±3.0 | 83.5 ±0.5 | 48.2 ±3.4 | 68.0 ±0.2 |
| | JTT | 71.2 ±0.5 | 48.3 ±1.5 | 82.6 ±0.4 | 65.1 ±1.6 | 64.9 ±0.3 |
| | LfF | 75.0 ±0.7 | 53.0 ±4.3 | 72.3 ±1.3 | 57.3 ±5.7 | 62.2 ±2.4 |
| | ○LISA | 77.5 ±0.7 | 57.8 ±0.8 | 79.0 ±0.8 | 66.8 ±0.3 | 66.8 ±0.6 |
| Imbalanced learning | ReSample | 70.0 ±1.0 | 74.1 ±2.2 | 81.0 ±1.7 | 66.8 ±0.5 | 67.5 ±0.3 |
| | ReWeight | 71.9 ±0.6 | 69.6 ±0.2 | 83.1 ±0.7 | 64.2 ±1.9 | 67.0 ±0.4 |
| | SqrtReWeight | 71.0 ±1.4 | 66.9 ±2.2 | 82.6 ±0.4 | 63.8 ±2.4 | 68.0 ±0.4 |
| | CBLoss | 74.4 ±1.2 | 65.4 ±1.4 | 83.1 ±0.0 | 63.6 ±2.4 | 67.6 ±0.3 |
| | Focal | 71.6 ±0.8 | 56.9 ±3.4 | 81.0 ±0.4 | 62.4 ±2.0 | 68.7 ±0.4 |
| | LDAM | 70.9 ±1.7 | 57.0 ±4.1 | 83.6 ±0.4 | 65.5 ±0.8 | 66.6 ±0.6 |
| | BSoftmax | 74.1 ±0.9 | 69.6 ±1.2 | 82.6 ±0.4 | 63.6 ±2.4 | 67.6 ±0.6 |
| classifier retraining | ●DFR | **89.0** ±0.2 | 73.7 ±0.8 | 81.4 ±0.1 | 63.8 ±0.0 | 67.1 ±0.4 |
| | CRT | 76.3 ±0.8 | 69.6 ±0.7 | 83.1 ±0.0 | 65.4 ±0.2 | 68.1 ±0.1 |
| | ReWeightCRT | 76.3 ±0.2 | 70.7 ±0.6 | 85.1 ±0.4 | 65.2 ±0.2 | 67.9 ±0.1 |

### 4.2.2 RESULTS AND DISCUSSION

We report the worst-group accuracy results obtained after running our FairDropout method on the subpopulation shift library, averaged over 5 independent runs. Table 1 presents these results with methods categorized according to the presentation done in Sec. 4.2.1, following Yang et al. (2023). As a reminder, in this setting, the spurious attribute and the group annotations are unavailable in both the training and validation datasets. All the methods or their adapted version are tuned with worst-class accuracy and the results come from Yang et al. (2023).

From the table, we can make the following observations. In all datasets and except Waterbirds, our FairDropout method outperforms ERM by a large margin.

On these datasets, we can also observe that the FairDropout outperforms or has comparable performance to spurious correlation methods and imbalance learning methods. More specifically, the datasets on which FairDropout achieves the most successful results are MultiNLI ($70.3 \pm 2.4$) and MIMIC-CXR ($70.6 \pm 0.6$).

On CelebA and MetaShift, although our FairDropout technique outperforms spurious correlation methods, its performance is comparable with the Resample on CelebA ($75.6 \pm 2.1$ vs $74.1 \pm 2.2$) and ReWeightCRT on Metashift ($85.9 \pm 1.1$ vs $85.1 \pm 0.4$). It is worth mentioning that our models with the FairDropout are trained with classic cross-entropy, meaning that the performance of our FairDropout technique can be further boosted with any of these existing imbalanced learning or classifier retraining methods.

On the Waterbirds dataset, although our FairDropout improves upon ERM, it underperforms classifier retraining methods and some imbalanced learning methods. Since Waterbirds is a dataset *synthetically* generated by placing bird objects into different backgrounds, it has already been observed that ImageNet pre-trained ImageNet features can be effectively transferred (Izmailov et al., 2022) without finetuning the entire model, which may explain the superior performance of DFR.

Overall, FairDropout proves to be an effective method for reducing reliance on spurious correlations without explicit group annotations. It may also benefit from additional boosts if combined with classifier retraining or imbalanced learning methods.

## 5 LIMITATIONS AND CONCLUSION

In this paper, we explored for the first time the lack of generalization of minority-group examples and its link to memorization in spurious correlation. We introduced the FairDropout, an example-tied dropout technique that can be applied in larger networks to reduce the reliance on spurious correlation.

FairDropout makes it possible to localize memorization and attracts the spurious features in such fixed neurons that once dropped during inference, can improve worst-group accuracy. We show empirical evidence that the FairDropout outperforms several baseline methods on datasets from image, medical, and language tasks.

However, our study has some limitations that have not been addressed. First, there have been works showing that there may exist memorization that is beneficial for generalization (Feldman, 2020)– this warrants further investigation, particularly in the case of the Waterbird dataset. Second, while implicitly by construction, we hypothesize that *generalizing* neurons are less likely to memorize examples since memorization is more easily achieved in the memorizing neurons, this assumption requires further exploration, which falls outside the scope of this paper.

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

Table 2: Hyperparameter ranges. Here dp$i$ stands for the position before the residual layer block $i$, dplogits stands for the position before the linear classifier head, dpfc stands for a position before the classifier head but after a newly introduced linear projection layer.

| Hyperparameters | sets |
|---|---|
| learning rate | $\{1e-3, 1e-4, 1e-5\}$ |
| Weight decay | $\{1e-3, 1e-4, 1e-5, 1e-6\}$ |
| $p_{\text{fixed}}$ | $\{.2, .3, .4, .5, 6\}$ |
| $p_{\text{mem}}$ | $\{.001, .1, .2, .4\}$ |
| FairDropout positions on ResNet-50 | $\{dp2, dp3, dp4, dp5\}$ |
| FairDropout positions on BERT | $\{dplogits, dpfc\}$ |

## A    APPENDIX

### A.1    HYPERPARAMETERS

Table 2 describes the range of hyperparameters that we used to tune the hyperparameters.

### A.2    MORE DETAILS ON THE DATASETS

Table 3: Dataset overview with data types, number of attributes, classes, train, validation, and test set sizes, and group distributions.

| Dataset | Data | $|\mathcal{A}|$ | $|\mathcal{Y}|$ | $|\mathcal{D}_{\text{tr}}|$ | $|\mathcal{D}_{\text{val}}|$ | $|\mathcal{D}_{\text{test}}|$ | Max group (%) | Min group (%) |
|---|---|---|---|---|---|---|---|---|
| Waterbirds | Image | 2 | 2 | 4795 | 1199 | 5794 | 3498 (73.0%) | 56 (1.2%) |
| CelebA | Image | 2 | 2 | 162770 | 19867 | 19962 | 71629 (44.0%) | 1387 (0.9%) |
| MetaShift | Image | 2 | 2 | 2276 | 349 | 874 | 789 (34.7%) | 196 (8.6%) |
| MultiNLI | Text | 2 | 3 | 206175 | 82462 | 123712 | 67376 (32.7%) | 1521 (0.7%) |
| MIMIC-CXR | X-rays | 6 | 2 | 303591 | 17859 | 35717 | 68575 (22.6%) | 7846 (2.6%) |

