# OpenReview forum: "FairDropout: Using Example-Tied Dropout to Enhance Generalization for Minority Groups"
_ICLR.cc/2025/Conference — ICLR 2025 Conference Withdrawn Submission_

### Official Review · Reviewer_MLGC · 2024-10-28

**Soundness:** 3
**Presentation:** 1
**Contribution:** 2
**Rating:** 5
**Confidence:** 3

**Summary:**

The main objective of this paper is to increase the performance of models that deal with spurious features. The authors firstly show that the poor performance on minority groups is due to the overfitting. They then use the recent works that connect generalization and memorization to propose FairDropout - algorithm that redirects this memorization to specific neurons that are subsequently dropped out during inference.

**Strengths:**

1. The problem description (3.1) is well written and allows for a good introduction to the topic of spurious correlations.
2. The idea is interesting and feels novel in comparison to the relevant literature. FairDropout incorporates the notion of memorization into the field of spurious correlations very nicely, opening paths to new research directions, as highlighted by the authors in Sec. 5.
3. The benchmarking is well described and the empirical results are great.

**Weaknesses:**

1. The writing is really poor. There are numerous typos, grammatical errors, and other language errors. The work requires major rewriting to be fit for a venue like ICLR. For example L98-L100, L142-L143, L145-L146, L189, L200. I suggest to the authors to carefully reread their paper and fix all of the glaring language errors, while also trying to simplify and clarify some of the more crucial sentences (for example, L178-L180 and L197-L200). The authors are almost 2 pages away from the page limit so they have a lot of space to improve the clarity of their work.
2. The description of groups in the introduction was very poor. They were, however, really neatly defined later on in Section 3.1. I expect the Introduction to be clear and accessible to people outside of the field of spurious features.
3. Figure 1 and 2 are crucial for validating the motivation behind FairDropout - poor performance on minority groups due to generalization/memorization. However, the description and analysis of the figures is lacking in clarity in my opinion. For Fig. 1, there is a mismatch between L189 and its labels, and it’s incorrectly referenced as Fig 2. In L189. For Figure 2b the description doesn’t feel clear to me. It is unclear to me how the Figures were obtained and what they represent. I expect the authors to clearly state what is the difference between minority group and worst-group (both definition-wise and practically on CellebA - which is which), and to clearly describe what they refer to as probability and how its computed.
4. The work of Maini et al. (2023) is crucial for this work, but I find the way in which it’s described (in L197-L210) very unintuitive and hard to understand. Furthermore, FairDropout is a modification of example-tied dropout from Maini et al., and yet no description nor intuition on example-tied dropout is introduced through the paper. I expect the authors to clearly state what example-tied dropout is and how it differs from their FairDropout. I also expect them to provide a clear and intuitive description of Eq. 1 so that the reader doesn’t have to read Maini et al. before reading this paper.

# Summary
Based on the results and relevant literature, I believe that the idea behind the work is novel in the field of spurious features. I really like the chain of thought in this paper - starting from the problem of spurious features, to identifying overfitting on minority groups and using the recent work, that connected memorization with generalization, to propose an elegant solution. My main issue with the paper is its presentation, and I am not sure if a work that was written so poorly should get accepted to a venue like ICLR. Furthermore, due to poor comparison with the method of Maini et al., I am not sure about the novelty of the proposed methodology. Is FairDropout something new or a simple rebranding of example-tied dropout that is applied in a new field? Due to that it's hard for me to evaluate the novelty of this work - it appears novel in spurious features field, but I am unsure if it's novel in broader context.

**Questions:**

1. Isn’t equation (1) an oversimplification assuming that the performance on all groups, apart from the worst-group, is satisfactory?
2. I don’t understand the description of FairDropout in L250-L260. Could authors help me out?
3. I don’t understand Figure 5 nor the caption. What are the coloured blocks next to the images? Why are they only connected with memorizing features?
4. Why do we care about training (and not testing) worst-group accuracy (Fig 2b & L215)?

---

### Official Review · Reviewer_pA9u · 2024-11-04

**Soundness:** 2
**Presentation:** 1
**Contribution:** 1
**Rating:** 3
**Confidence:** 5

**Summary:**

The paper aims to mitigate models' reliance on spurious features without requiring group annotations in the training set.
Building on recent findings that memorization can be localized to a limited number of neurons, the paper uses example-tied
dropout, termed FairDropout, to redirect the memorization on spurious features to specific neurons that are subsequently dropped out during inference. Experiments demonstrate the effectiveness of this proposed method.

**Strengths:**

The paper focuses on mitigating models' reliance on spurious correlations in a practical setting where no group annotations are available in the training and validation sets. Thus, the proposed method can be very useful in a wide range of settings where group annotations are hard to acquire.

**Weaknesses:**

- The technical contribution of this paper is limited. The proposed method is a direct follow-up of the method in [1].

- The paper is poorly written. Fixing typos (e.g., Line 189: Fig. 2 -> Fig. 1, and Line 145:$p_{tr}$), grammar errors (e.g., Line 161: there no -> there is no), and format issues (e.g., different font sizes in Fig. 1) would be a good starting point. More importantly, the method is not clearly described in the paper. Specifically, how to determine what neurons represent generalizing or memorizing features is unclear.

- The proposed method needs to determine memorization neurons for each training example, which may be prohibitive when the dataset is large and there are many neurons in a model's internal layer. It would be helpful to provide the training time or an analysis on the computational complexity.

- The experiments lack a comparison with some recent methods, e.g., [2] and [3]. Moreover, the reported performance of existing methods is not standard, making it difficult to assess the effectiveness of the proposed method, in comparison with other methods such as [2] or [3]. The effectiveness of the proposed method may be strengthened if the method is evaluated in the setting of [2] or [3].

[1] Maini et al., Can neural network memorization be localized? ICML, 2023.\
[2] LaBonte et al., Towards last-layer retraining for group robustness with fewer annotations, NIPS, 2023.\
[3] Li et al., Bias Amplification Enhances Minority Group Performance, TMLR, 2024.

**Questions:**

Without group annotations, how is it possible to determine whether a neuron represents a memorizing feature or a generalizing feature?

---

### Official Review · Reviewer_tm8z · 2024-11-04

**Soundness:** 2
**Presentation:** 1
**Contribution:** 2
**Rating:** 3
**Confidence:** 3

**Summary:**

The authors propose a dropout-based model to tackle spurious correlations. They first perform an analysis similar to Maini et al. (2023), showing empirically on CelebA that neural networks tend to memorize minority training examples. Motivated by this finding, they adapt the example-tied dropout method proposed by Maini et al. (2023) to improve worst-group accuracy. In this approach, "dropout layers" are added to the network. In such a layer, some fraction of neurons (designated for generalization) are never dropped out, whereas the remaining neurons (designated for memorization) are uniformly assigned across training examples, activated only when the corresponding example is seen, and completely dropped out at test time. The authors evaluate their approach on the benchmark suite in Yang et al. (2023).

[1] Maini, Pratyush, et al. "Can neural network memorization be localized?" (2023)

[2] Yang, Yuzhe, et al. "Change is hard: A closer look at subpopulation shift." (2023)

**Strengths:**

1. The memorization analysis in Section 3.2 is illuminating and provides further evidence that ERM overfits to noise in minority training examples in order to improve training loss. This is the key mechanism by which spurious correlations are learnt and Figure 2 demonstrates this phenomenon quite well.

2. The evaluation in Section 4 is comprehensive and compares the author's approach to various other methods across a number of datasets. The authors neatly categorize these methods by the assumptions/additional data they require.

**Weaknesses:**

1. It is not entirely clear to me where the novelty of the paper lies.
    - It is well-known that ERM overfits to noise in minority training examples. This is how spurious correlations are learnt and evidence for this has been previously shown already, e.g. by works like Puli et al. (2023) that the authors themselves have cited. As such, the result in Section 3.2 showing that ERM memorizes training minority samples is helpful but not novel or surprising.
    - It is also unclear to me how FairDropout is distinct from example-tied dropout as first proposed by Maini et al. (2023). I could be misunderstanding something, but besides the minor detail that dropout layers can also be applied to the projection heads, the actual implementation seems identical to example-tied dropout. As far as I can tell, it shares the same hyperparameters $p_{gen}$ and $p_{mem}$, etc.

2. The analysis in Section 3.2 is limited to CelebA. While CelebA is a benchmark for spurious correlations, it is a narrow dataset confined to faces and as such, a relatively easy dataset. I am not immediately convinced that conclusions made about memorization in CelebA can be generalized to harder datasets. At least, it would be good to show this same analysis for Waterbirds or ImageNet.

3. It is not immediately clear that deploying dropout in this manner actually fixes spurious correlations. Is there any guarantee or reasoning that generalization neurons are not also memorizing training examples, or (conversely) that memorization neurons don't learn generalizable features? The original work by Maini et al. was not designed to tackle spurious features, so I would like to some stronger evidence that this key assumption holds true. For example, at the very least, we should be able to replicate the analysis in Section 3.2 to show that zeroing out generalization/memorization neurons do/do not flip the prediction of majority/minority examples.

4. I suspect that this approach --- much like other assumption-based methods that don't require group labels (e.g. JTT, Liu et al. (2021)) --- will be very sensitive to hyperparameters. Is there: (a) any ablation for how results vary across hyperparameters, or (b) some discussion about what good hyperparameters might look like? In particular, it is a bit unclear to me how many and where the dropout layers should be placed throughout the model.

5. Finally, the actual results are a little mixed. Dropout does not seem to do well on a more challenging benchmark like Waterbirds. It has stronger results on CelebA and MultiNLI but only incremental gains on MetaShift and CXR.

[1] Puli, Aahlad Manas, et al. "Don’t blame dataset shift! shortcut learning due to gradients and cross entropy." (2023)

[2] Liu, Evan Z., et al. "Just train twice: Improving group robustness without training group information." (2021)

**Questions:**

I would like the authors to chiefly respond to the points raised in the Weaknesses section. There are also some minor questions/comments below:

- L189: I think you mean to refer to Figure 1 in this paragraph, not Figure 2.
- Fig 4: For the left-most image, which mode (train or test) was it generated under?

P.S. There are a bunch of typos and some \citep and \citet issues. I would encourage the authors to give the paper a once-over at some point in the future.

---

### Official Review · Reviewer_vC67 · 2024-11-05

**Soundness:** 2
**Presentation:** 2
**Contribution:** 2
**Rating:** 3
**Confidence:** 5

**Summary:**

This paper proposes a group-label-free method for robustness against spurious correlations in deep neural networks, based on the observation that the models generalize well on majority groups but memorize samples from minority groups. The authors propose a method, FairDropout to dropout certain neurons (which memorize minority samples) during inferences. The experiments on five widely used datasets show the effectiveness of the proposed methods in worst-group accuracies.

**Strengths:**

1. The structure of this paper is clear and easy to follow.
2. The paper proposes a practical method to mitigate the reliance on spurious correlations without any group annotations. This is a promising direction to better robustness.
3. The method does not introduce additional computational resources while effectively mitigating spurious correlations.

**Weaknesses:**

1. Missing several key baseline methods [1,2,3] in the same settings (without group labels). The reported worst-group accuracies are lower than those of these methods, which limits the contribution.
2. The observation lacks an in-depth and theoretical analysis. The observation may not always hold when it is in an under-parameterization or over-parameterization regime. See section 5.3 in [4]. To empirically validate the idea, experiments with different backbones (e.g. ResNet-101, ViT) are required or you can provide a theoretical analysis (or both).
3. Eq. 2 is hard to understand. What does the gradient subscripting $j$ mean? Section 3.2 needs to be revised for the audience to understand how certain neurons are identified.
4. There lacks an ablation study on $p_{\text{gen}}$ and $p_{\text{mem}}$.


[1] Simple and Fast Group Robustness by Automatic Feature Reweighting, ICML 2023 \
[2] Bias Amplification Enhances Minority Group Performance, TMLR 2024 \
[3] Correct-n-Contrast: A Contrastive Approach for Improving Robustness to Spurious Correlations, ICML 2022 \
[4] An investigation of why overparameterization exacerbates spurious correlations, ICML 2020

**Questions:**

- See the weaknesses.
- Can you explain why the results of Waterbirds are the worst among the provided baseline methods, considering it is commonly known as the easiest dataset to learn? Line 425-429 does not seem to give a proper explanation.
- Line 323: "Code is available at this ANONYMOUS LINK." The link is not provided.

---

### Note · Authors · 2024-11-15

**Comment:**

We thank the reviewers for their time and constructive feedback. We will incorporate these suggestions into the revised version of our paper.

**Withdrawal Confirmation:**

I have read and agree with the venue's withdrawal policy on behalf of myself and my co-authors.